# Ear, Nose and Throat (ENT) disease diagnostic error in low-resource health care: Observations from a hospital-based cross-sectional study

**Lufunda Lukama**[1,2]*, **Colleen Aldous**[2], **Charles Michelo**[3,4], **Chester Kalinda**[5,6]

1 Department of Otorhinolaryngology, Head and Neck Surgery, Ndola Teaching Hospital, Ndola, Zambia, 2 College of Health Sciences, Nelson R Mandela School of Clinical Medicine, University of KwaZulu-Natal, Durban, South Africa, 3 School of Public Health, Department of Epidemiology, Harvest University, Lusaka, Zambia, 4 Strategic Centre for Health Systems Metrics & Evaluations (SCHEME), School of Public Health, University of Zambia, Lusaka, Zambia, 5 Bill and Joyce Cummings Institute of Global Health, University of Global Health Equity, Kigali, Rwanda, 6 Howard College Campus, College of Health Sciences, School of Public Health and Nursing, University of KwaZulu-Natal, Durban, South Africa

\* lufundal@gmail.com

**Data Availability Statement:** The data used in this study can be accessed here: Lukama, Lufunda; Aldous, Colleen; Michelo, Charles; Kalinda, Chester (2022) "Dataset on diagnostic errors of Ear, Nose

## Abstract

Although the global burden of ear, nose and throat (ENT) diseases is high, data relating to ENT disease epidemiology and diagnostic error in resource-limited settings remain scarce. We conducted a retrospective cross-sectional review of ENT patients' clinical records at a resource-limited tertiary hospital. We determined the diagnostic accuracy and appropriateness of patient referrals for ENT specialist care using descriptive statistics. Cohens kappa coefficient (κ) was calculated to determine the diagnostic agreement between non-ENT clinicians and the ENT specialist, and logistic regression applied to establish the likelihood of patient misdiagnosis by non-ENT clinicians. Of the 1543 patients studied [age 0–87 years, mean age 25(21) years (mean(SD)], non-ENT clinicians misdiagnosed 67.4% and inappropriately referred 50.4%. Compared to those aged 0–5 years, patients aged 51–87 years were 1.77 (95%CI: 1.03–3.04) fold more likely to have a referral misdiagnosis for specialist care. Patients with ear (aOR: 1.63; 95% CI: 1.14–2.33) and those with sinonasal diseases (aOR: 1.80; 95% CI: 1.14–2.45) had greater likelihood of referral misdiagnosis than those with head and neck diseases. Agreement in diagnosis between the ENT specialist and non-ENT clinicians was poor (κ = 0.0001). More effective, accelerated training of clinicians may improve diagnostic accuracy in low-resource settings.

## Introduction

Diseases affecting the ear, nose and throat (ENT) constitute 20–50% of disorders treated at health facilities [1]. In most resource-limited settings, they are associated with poor outcomes [2]. Despite their high prevalence, ENT diseases have been disregarded in global health [3]. In Zambia and other developing countries, ENT health care is poorly funded and has inadequate

and Throat (ENT) diseases ", Mendeley Data, v1 (http://dx.doi.org/10.17632/6tgz2db7yn.1).

**Funding:** The authors received no specific funding for this work. However, during the course of this study, Lufunda Lukama received a tuition fee waiver from the University of KwaZulu-Natal (https://ukzn.ac.za) and a scholarship (tuition fees, living cost stipend) from the Canon Collins Educational and Legal Assistance Trust (https://canoncollins.org/) for his Doctor of Philosophy studies at the University of KwaZulu-Natal College of Health Sciences in South Africa [no grant numbers available]. There was no additional external funding received for this study. The funders had no role in study design, data collection and analysis, decision to publish, or preparation of the manuscript.

**Competing interests:** The authors have declared that no competing interests exist.

infrastructure, equipment, medication, human resources and training facilities [4, 5]. The few available ENT specialists are distributed to limited overwhelmed urban units. Thus, most ENT patients are treated by clinicians not trained in ENT disease management [4]. Although these clinicians must be competent in the basic management of ENT conditions, many countries have deemed them inadequately trained to treat ENT diseases [6]. As a result, the true incidence of diagnostic error (delayed, missed or wrong diagnosis) [7] and inappropriate or late ENT patient referrals for treatment among non-ENT clinicians is likely high.

Diagnostic error contributes up to 70% of all medical errors [8]. In well-resourced countries, the incidence of diagnostic error varies between 0.7 to 15%, of which half is harmful [9, 10]. However, in low-resource settings, data relating to harmful diagnostic error remains sparse due to limited access to diagnostic resources, shortage of qualified medical professionals and poverty of electronic record-keeping systems [11]. While inappropriate referral of patients can undermine the quality of patient care [12], misdiagnosis increases the risk of death, elevates the financial strain on the already stretched health systems and may result in significant litigation [13]. In Zambia, there has not been an investigation into the prevalence and impact of diagnostic errors in health facilities.

In this study, we determined the diagnostic accuracy and estimated the appropriateness of referrals among patients treated at Zambia's highest ENT treatment facility. In addition, we determined the level of agreement of ENT diagnoses between the ENT specialist and referring clinicians.

## Methods

### Design and setting

We conducted a retrospective cross-sectional review of clinical case records of patients presenting to the University Teaching Hospitals (UTH) ENT outpatient clinic in Zambia between 1 July 2019 and 30 October 2019. Patients were referred by non-ENT clinicians from different health facilities in other regions of the country. A more comprehensive retrospective review was not practical as most of the patients' records that were supposed to be filed before July 2019 and later than October 2019 could not be traced due to poor record-keeping, a general problem noted in Zambia [14]. The trial filing system used between July 2019 and October 2019 improved patient record-keeping in the absence of an electronic filing system.

The UTH complex of hospitals is Zambia's highest referral health facility, with its only ENT clinic receiving referrals from the country's 10 provinces of 18.5 million people [15]. Until August 2019, it was the country's only public health facility with an ENT specialist. In Zambia, the patient referral pathway ascends from health centres, clinics, level 1 (first level) hospitals, level 2 (second level) hospitals and level 3 (third level) hospitals. Level 1 hospitals manage the most basic, level 2 hospitals intermediate-level and level 3 hospitals the most complex medical conditions [16]. Clinics and level 1 hospitals have basic laboratory and imaging equipment (ultrasound, x-ray) while higher-level hospitals have advanced diagnostic facilities.

### Participant selection and sampling

Initially, using hardcopy registers, we identified and retrieved files of all 1,873 patients presenting to the facility within the study period. A two-phase screening process followed, in which records to be included in the final analysis were isolated. Phase 1 excluded all duplicates, leaving 1,701 files. In phase 2, we screened the 1,701 medical records for completeness, including the dates of presentation and referral information (referral centre, diagnosis, and reason for referral). We excluded records with missing referral information and used the remaining 1,543 records for analysis.

## Measurements and outcomes

Our study outcome measures were the patient's non-ENT clinician (referral) diagnosis and that of the ENT specialist. The considered ENT specialist diagnosis was that made after investigating the patient as applicable (definitive or final diagnosis). We considered similar diagnoses 'matching' and marked missing diagnoses 'unknown' because we assumed that the clinician did not know it by not assigning a diagnosis to the patient. Queried diagnoses made by the referring clinicians and the ENT specialist were assumed definite because they often fall within the broad category of the actual patient pathology, as has been the principal researcher's experience. With further assumption that the ENT specialist diagnoses were correct, patient referral diagnoses which did not match the ENT specialists' were classified '*misdiagnoses*'. The International Classification of Disease 11th Revision (ICD11) was used for ENT diagnoses whenever appropriate. For this study, we defined 'inappropriate referral' according to Blundel et al. [17]. A referral was inappropriate if it was either unnecessary (all available primary care options had not been exhausted), and/or had a wrong referral destination (patient referred to ENT instead of another medical speciality), and/or was of poor quality (necessary tests and investigations were not performed considering the available resources of the referring facility).

## Data collection and statistical considerations

For data abstraction and collection, we utilized Microsoft excel soft copies of a standardized data collection tool (S1 Appendix). We recorded the patient's demographic and clinical characteristics, including the patient's date of the hospital visit, age, sex, town, and province of residence and that referred from, referring facility and its level of care, referral diagnosis, the reason for referral, ENT specialist diagnosis and whether the patient was a first attendant, had new pathology or was for review.

We analysed the data using STATA version 15.0 (STATA Corp, Texas, USA). Descriptive statistics (frequencies, percentages) were used to summarise categorical data and describe patient socio-demographic characteristics, referral and ENT specialist diagnoses, epidemiological disease profiles and prevalence of misdiagnosis and inappropriate patient referrals. Using bivariate analysis, we explored the association between socio-demographic characteristics and referral hospital characteristics with the diagnosis made and the appropriateness of the referral. We used univariate and multivariate logistic regression to determine factors that influenced the clinician diagnosis and appropriateness of referral. Statistical significance was defined by $p < 0.05$. In addition, we determined the level of agreement of the referring facility diagnosis and that made by the ENT specialist at UTH using the Cohens kappa coefficient ($\kappa$). This manuscript was written using the STROBE checklist for cross-sectional studies [18] (S2 Appendix).

## Ethics

The protocol and procedures of this study were approved by the University of KwaZulu-Natal Biomedical Research Ethics Committee (Ref: BREC/00001984/2020) following the South African national guidelines on Biomedical Research. In addition, ethics approval was obtained from the University of Zambia Biomedical Research Ethics Committee (Ref: REF. NO. 1380–2020) and the Zambia National Health Research Authority (Ref: NHRA00007/15/03/2021). Informed consent was waived by the Ethics Boards as the research was of minimal risk to the patients. Gatekeeper permission to conduct the study was obtained from UTH [no Reference number]. All data was fully anonymised following completion of the study. The study was cleared for publication by the Zambia National Health Research Authority (NHRA).

## Results and analysis

### Patient demographics

1543 patients of age 0–87 years (mean age 25 ± 21 years [mean ± SD]) were included in the study, comprising 47.3% (n = 729) children (≤18 years of age) and 52.7% (n = 814) adults. The male: female patient ratio was 1:1.1 (725 males and 818 females). 88.9% (n = 1372) were health facility referrals while 11.1% (n = 171) were self-referrals.

### ENT disease epidemiology

Head and neck diseases (34.8%, n = 537) were the most predominant, followed by sinonasal (34.4%, n = 530), and ear (25.7%, n = 397) diseases (S1 Table). The most frequently treated diseases were acute pharyngotonsillitis (9.5%, n = 146), allergic rhinitis (9.1%, n = 140), chronic/recurrent pharyngotonsillitis (7.8%, n = 121), adenoid hypertrophy (6.4%, n = 99) and chronic otitis media (COM) [6.3%, n = 97].

### Diagnosis accuracy and mismatch

Overall, 67.4% (n = 916) of all referral diagnoses did not match the ENT specialists'. Most (36.5%, n = 333) of the misdiagnosed patients had sinonasal diseases and the least (26.2%, n = 239) had ear diseases. Further, the majority (25.1%, n = 230) of misdiagnosed patients were aged 0–5 years, while the minority (4.4%, n = 40) were aged 13–18 years. Most (74.1%, n = 80) misdiagnoses were from level 1 hospitals (S2 Table). Despite non-ENT clinicians correctly diagnosing 72.0% (n = 18) of nasal polyps and 51.5% (n = 35) of acute tonsillitis in patients referred for specialist care, only 3.3% (n = 4) of them correctly diagnosed allergic rhinitis. Table 1 outlines the common diagnoses made by the ENT specialist at UTH ENT clinic, and proportions of their corresponding referring (non-ENT) clinician diagnoses.

In the multivariate logistic regression model, patients aged 51–87 years were 1.77 (aOR: 1.03–3.04) fold more likely to be misdiagnosed by the referring clinician than those aged 0–5 years. Patients with ear diseases (aOR: 1.63; 95% CI: 1.14–2.33), those with sinonasal diseases (aOR: 1.80; 95% CI: 1.14–2.45) and those with no ENT pathology (aOR: 3.08; 95% CI: 2.58–4.48) were more likely to have a wrong pre-ENT referral diagnosis compared to those with head and neck diseases (Table 2).

### Appropriateness of referrals

Of the total 1372 non-ENT specialist clinician referrals, 56.6% (n = 777) were inappropriate. Of these, 67.3% (n = 523) could have been treated by non-ENT clinicians at lower-level hospitals, 23.9% (n = 186) required trial of medical treatment before referral to UTH and 7.2% (n = 56) had no indication for ENT treatment. Patients aged 0–5 years were the most (27.4%, n = 213), while those aged 13–18 years were the least (4.6%, n = 36) inappropriately referred. The majority (39.1%, n = 302) of inappropriate referrals had head and neck diseases, while the minority (0.4%, n = 3) had an ENT-unrelated medical problem (S3 Table). Univariate and multivariate logistic regression showed that referrals from within the UTH complex of hospitals were 1.35 (95% CI: 1.03–1.75) and 1.34 (95% CI: 1.03–1.75) more likely to be inappropriate than those from other facilities (Table 3).

### Diagnosis agreement

The overall agreement between the diagnoses made by the referring clinicians and those of the ENT specialist was low (κ = 0.0001) [p = 0.473], as were those between Level 1 hospital referring clinicians and ENT specialist (κ = 0.0201) [p = 0.046], between Level 2 hospital referring

**Table 1. Disease diagnosis comparison between ENT specialist and non-ENT clinicians.**

| ENT specialist diagnosis n* | Non-ENT clinician diagnosis of the disease Disease, n (%) | | | | |
|---|---|---|---|---|---|
| *Allergic rhinitis 120* | No diagnosis indicated 38 (31.7) | Sinusitis 19 (15.8) | Nasal polyps# 15 (12.5) | Adenoid hypertrophy 9 (7.5) | Nasal tumour 5 (4.2) | Allergic rhinitis 4 (3.3) |
| *Adenoid hypertrophy 90* | No diagnosis indicated 34 (37.7) | Adenoid hypertrophy 17 (18.9) | Tonsillitis 17 (18.9) | Nasal polyps 6 (6.7) | palatine tonsillar hypertrophy 4 (4.4) | Sinusitis 3 (3.3) |
| *Chronic otitis media 87* | Chronic otitis media 37 (42.5) | No diagnosis indicated 22 (25.3) | Hearing loss, unspecified cause 11 (12.6) | Acute otitis media 7 (8.0) | Others 10 (11.5) (each with n = 1, 1.1%) | - |
| *Acute tonsillitis 68* | Acute tonsillitis 35 (51.5) | Adenoid or tonsillar hypertrophy 8 (11.8) | Chronic tonsillitis 7 (10.3) | No diagnosis indicated 6 (8.8) | Acute pharyngitis 4 (5.9) | Others 8 (11.8) (each with n = 1, 1.5%) |
| *Laryngo-pharyngeal reflux 47* | No diagnosis indicated 17 (36.2) | Chronic pharyngo-tonsillitis 7 (14.9) | Acute pharyngo-tonsillitis 7 (14.9) | Laryngo-pharyngeal reflux 2 (4.3) | Laryngeal tumour 2 (4.3) | Laryngeal tumour 2 (4.3) |
| *Ear wax occlusion 46* | Hearing loss, unspecified cause 18 (39.1) | No diagnosis indicated 11 (23.9) | Otitis media 8 (17.4) | Ear wax impaction 6 (13.0) | Foreign body ear 2 (4.3) | Sinusitis 1 (2.3) |
| *Laryngeal papilloma 42* | No diagnosis indicated 23 (54.8) | Laryngeal papilloma 5 (11.9) | Chronic laryngitis 3 (7.1) | Tracheal tumour 2 (4.8) | Others 9 (21.4) (each with n = 1, 2.4%) | - |
| *Acute otitis media 34* | No diagnosis indicated 15 (44.1) | Acute otitis media 12 (35.3) | Others 7 (20.6) (each with n = 1, 2.9%) | - | - | - |
| *Chronic rhinosinusitis without nasal polyps 32* | No diagnosis indicated 9 (28.1) | Chronic rhinosinusitis 9 (28.1) | Nasal polyps 4 (12.5) | Recurrent tonsillitis 2 (6.3) | Others 8 (25.0) (each with n = 1, 3.1%) | - |
| *Chronic rhinosinusitis with nasal polyps 25* | Nasal polyps 18 (72.0) | No diagnosis indicated 3 (12.0) | Chronic rhinosinusitis 2 (8.0) | Nasal tumour 1 (4.0) | Acute sinusitis 1 (4.0) | - |

* Total count (n) excludes the number of patients who referred themselves for ENT specialist care (self-referrals) and therefore, did not have a pre-referral non-ENT clinician diagnosis

# Patients had no nasal polyps on clinical evaluation by the ENT specialist

clinicians and ENT specialist ($\kappa$ = -0.0214) [p = 0.870] and between Level 3 hospital referring clinicians and ENT specialist ($\kappa$ = 0.0020) [p = 0.869]

## Discussion

In this study, we determined that the diagnostic error and inappropriate patient referrals for specialist ENT care in our resource-limited setting were high. Our findings provide evidence of and highlight the need to improve the diagnostic accuracy among medical doctors to improve the quality of ENT patient care. Although ENT is a surgical speciality, the skills it encompasses are relevant to all medical and surgical disciplines [19]. Therefore, all doctors are required to be competent to diagnose and treat basic ENT diseases and refer patients appropriately [19].

Worldwide, most health-care visits by patients are due to ENT diseases, most of which relate to the ear [20]. In contrast, this study determined that the majority (34.8%) of patients attending the UTH ENT clinic had diseases affecting the head and neck and the minority (25.7%) had those affecting the ear. This finding was inconsistent with evidence from the developing world, including India [21] and Nigeria [22], where ear diseases form the majority of ENT diseases treated in health facilities. While bigger multi-centre studies may be required to verify this finding, a possible explanation could be that many people with ear diseases (e.g., hearing loss, chronic otitis media) in Zambia do not readily access ENT services due to poor socioeconomic conditions, unavailable ENT services in health care facilities and poor health-

**Table 2. Univariate and multivariate logistic regression of the pre-ENT referral misdiagnosis among the different patient groups.**

| | Univariate Analysis | | | Multivariate Analysis | | |
|---|---|---|---|---|---|---|
| Variable | OR | p-value | 95% CI | aOR | p-value | 95% CI |
| 0–5 years | Reference | | | | | |
| 6–12 years | 0.98 | 0.887 | 0.69–1.38 | | | |
| 13–18 years | 0.73 | 0.218 | 0.44–1.21 | | | |
| 19–35 years | 1.34 | 0.071 | 0.98–1.84 | | | |
| **36–50 years** | **1.79** | **0.003** | **1.22–2.62** | | | |
| **51–87 years** | **2.02** | **0.001** | **1.34–3.02** | **1.77** | **0.039** | **1.03–3.04** |
| Departments: Medicine | Reference | | | | | |
| Paediatric | **0.38** | **0.001** | **0.25–0.57** | **0.53** | **0.012** | **0.3–0.87** |
| Surgery | **0.60** | **0.013** | **0.41–0.89** | | | |
| Women and Newborn | 0.14 | 0.115 | 0.01–1.61 | | | |
| ENT subspecialty | | | | | | |
| Head and Neck | Reference | | | | | |
| **Otology** | **1.802** | **0.001** | **1.34–2.43** | **1.63** | **0.008** | **1.14–2.33** |
| **Rhinology** | **1.691** | **0.001** | **1.29–2.21** | **1.80** | **0.001** | **1.32–2.45** |
| Medical problem | 0.537 | 0.419 | 0.12–2.43 | | | |
| **No ENT pathology** | **1.396** | **0.001** | **1.17–1.67** | **1.78** | **0.025** | **1.08–2.95** |

OR: Odds ratio; aOR: adjusted odds ratio; 95% CI: 95% Confidence interval

Note: multivariate logistic regression data omitted from the table were not significant at p < 0.05

seeking behaviour [4, 5, 23]. Considering the high (77.9%) poverty levels and large (56.4%) population of rural Zambia [15], and that ENT services are only found in few urban hospital units [4], many ear disease-affected patients in rural areas may not have insight into the potential adverse outcomes of these diseases to warrant long-distance travel to access an ENT service [24, 25]. Therefore, more effective universal health education, improved socioeconomic conditions and disadvantaged communities' improved access to ENT services may result in a different epidemiological outlook of ENT diseases from what has been observed in the current study.

Our results suggest a high prevalence of diagnostic error among non-ENT trained health workers in low-resource settings. While data comparing diagnostic error between medical and surgical subspecialties is scarce, available evidence suggests a lower prevalence of diagnostic error among physicians and paediatricians. For instance, a survey conducted on 583

**Table 3. Univariate and multivariate logistic regression of inappropriate referrals among the different patient groups.**

| | Univariate Analysis | | | Multivariate Analysis | | |
|---|---|---|---|---|---|---|
| Variable | OR | p-value | 95% CI | aOR | p-value | 95% CI |
| Attendance status | | | | | | |
| Review | Reference | | | | | |
| **First attendance** | **1.36** | **0.006** | **1.09–1.68** | **1.35** | **0.005** | **1.09–1.68** |
| Attendance not stated | 0.91 | 0.946 | 0. 06–1.46 | | | |
| Referral within UTH No | Reference | | | | | |
| Yes | **1.35** | **0.027** | **1.04–1.75** | **1.34** | **0.028** | **1.03–1.75** |

OR: Odds ratio; aOR: adjusted odds ratio; 95% CI: 95% Confidence interval

Note: multivariate logistic regression data omitted from the table were not significant at p < 0.05

physicians across the United States of America found diagnostic error of 4.5% [26]. Similarly, diagnostic error of 3.6% among primary care physicians was observed in Malaysia [27]. In another study, more than half of paediatricians reported diagnostic error at least once or twice a month, and they made harmful errors at least once or twice a year [28]. Further, a study in the Netherlands reported diagnostic accuracy of 0.42 among medical residents, scored as either 0 (incorrect), 0.5 (partially correct) or 1 (correct) [29]. With 96.7% of allergic rhinitis, 95.7% of laryngopharyngeal reflux and 87.0% of ear wax impaction missed by non-ENT clinicians in our study, most of the patients with ENT conditions are likely inappropriately treated with the wrong medication and surgery. Further, some patients may not be referred early for management of life-threatening conditions like cancer. This may worsen the disease prognosis and increase the ultimate cost of patient care due to prolonged hospital stay and more advanced, costly treatment. In addition, delayed diagnosis of benign ENT conditions may result in life-threatening complications (airway obstruction in laryngeal papilloma, intracranial sepsis in acute otitis media and rhinosinusitis).

The poor agreement between the referring clinicians and ENT specialist ($\kappa = 0.0001$) and the high rate (56.6%) of inappropriate referrals for ENT specialist care may indicate inadequate training and poor knowledge of basic ENT disease management among most clinicians. In Zambia and the rest of the world, ENT is given little weight in medical school curricula, with most ENT undergraduate curricula offering a 1–2 week predominantly observational rotation [30]. As such, most clinicians lack the necessary skills to confidently manage ENT diseases upon completion of their undergraduate medical training [31]. Even if some literature emphasizes the superiority of adequate undergraduate ENT medical training over the duration spent as a general practitioner in the acquisition of competence to treat ENT conditions [32], the majority of clinicians often do require additional training to improve their competence following their undergraduate medical education [31]. Therefore, improving ENT medical education and clinical exposure at undergraduate, graduate and in-service training in Zambia would give clinicians better knowledge and skill to handle ENT conditions [32, 33]. A study assessing the knowledge, attitudes, and current practices of health workers with regard to the basic management of ENT diseases in Zambia is ongoing. We hope that the study will provide more insight into why some uncomplicated and correctly diagnosed conditions (i.e., acute tonsillitis, allergic rhinitis) are sent for ENT specialist treatment.

Literature has conflicting evidence on the rate of disease misdiagnosis in different age groups, with some reporting higher rates in older age groups [34] and some in younger ones [35]. In our study, patients aged 51–87 years were 1.8 times more likely to be misdiagnosed by the non-ENT specialist than those aged between 0–5 years. While we cannot provide a reason for this finding, it is worth exploring further with a more robust study with more participants. However, the widespread shortage of ENT diagnostic instruments across Zambia may explain the higher likelihood of misdiagnosis of patients with ear and those with sinonasal diseases. As nasal specula, otoscopes and endoscopes are seldom available in health facilities [5], clinicians seldomly perform complete ear and nasal examinations for diagnostic purposes. Except that of the larynx, much of the head and neck pathology requires no instrumentation to complete. In addition to the lack of sufficient skills and equipment at lower-level hospitals, patient referral networks in Zambia and many other low-income nations are undermined by poor coordination between lower and upper-level hospitals [36]. This may, in part, explain the high rate of inappropriate referrals noted in this study.

Even though debatable, available evidence shows that the knowledge and experience gained from regular practice of the clinician matters in reducing diagnostic errors, with more experienced clinicians likely making fewer diagnostic errors [37]. In our study, however, this trend was not apparent as patients referred from level 3 hospitals were not the least misdiagnosed

(66.8%) [even though those referred from level 1 hospitals were the most misdiagnosed (74.1%)]. This finding supports the evidence that doctors' medical knowledge stagnates after completing undergraduate training, but we suggest gathering better evidence with a more robust review of the literature. Our study also suggests that the referral practices in high-level hospitals (level 3) ought to be audited to ascertain whether junior doctors do some referrals to ENT without more experienced senior input, a practice that may elevate the rate of diagnostic error and inappropriate referral. In Zambia, level 1 hospitals are manned by Senior Resident Medical Officers (doctors who have completed at least 1.5 years of consultant-supervised practice following completion of undergraduate medical training but not yet commenced specialist training), while level 3 hospitals are training sites for medical students, Junior Resident Medical Officers (newly qualified doctors undergoing consult supervision and not fully registered as medical practitioners with the Health Professions Council of Zambia), as well as specialist trainees.

To make quality ENT services more available to the Zambian public, non-ENT specialist cadres can be trained to diagnose and manage basic ENT diseases, a strategy that has been successful in other countries within the region, such as Malawi [38]. Some literature from the developing world suggests that health workers can be trained to reliably diagnose and treat ENT diseases [39], with one study in Malawi [40] reporting good ($\kappa = 0.7$) and moderate ($\kappa = 0.5$) agreements in ear pathology diagnosis between the ENT specialist and the ENT clinical officer (mid-level health care providers with less training and more restricted scope of practice than medical doctors) and nurse respectively. The investigators noted the potential for non-specialist health workers to be involved in the patient assessment of ENT disease. Accordingly, training these non-specialist health workers in the use of basic ENT equipment (otoscopes, tuning forks, nasal specula), performance of basic procedures (ear syringing, indirect laryngoscopy) and non-expert interpretation head and neck x-rays would improve the quality of patient care in many places lacking an ENT specialist. In addition to training non-ENT specialists in ENT disease management, establishing international partnerships to train specialists locally would be successful in steadily increasing the pool of ENT specialists in Zambia. Such initiatives have been effective in advancing ENT in Africa, including The CBM International assistance in the establishment of ENT units in Central Africa, The University of Cape Town Karl Storz Head, Neck and Paediatric ENT Fellowships training African head, neck and paediatric ENT Fellows, and Operation Ear Drop Kenya equipping the temporal bone laboratory in Nairobi. Further, the introduction of ENT telemedicine to Zambia may improve patients' access to quality health care. While telemedicine has been successfully implemented in other medical subspecialties in Zambia [41], it is yet to be used in ENT health care. However, to ensure success of telemedicine, the government must develop its implementation strategies that are tandem with Zambia's needs [42].

The World Health Organisation (WHO) recognizes the importance of errors in diagnosis in its prioritization of safety errors in primary health care [43]. With up to 50% of GP consultations due to ENT conditions, the accuracy of diagnosis of ENT diseases remains a vital component of quality health care [43]. As serious harm rates of misdiagnosis per incident disease can be as high as 36% [44], all efforts to increase the ENT workforce in Zambia must be accompanied by effective mechanisms to guarantee knowledgeable and competent clinicians for ENT health care. While Cannon et al. (2004) estimated that one otolaryngologist for every 40,000 population is sufficient to provide comprehensive patient care [45], Zambia's 0.007/40,000 ENT specialist/population ratio is well below the proposed minimum requirement [4]. Considering that diagnostic errors are the largest proportion of medical errors (up to 70%) [8], lowering ENT clinicians' quality would be additional harm to the Zambian health system. While doctors often work towards correcting health-care system factors that lead to medical

error, they are reluctant to acknowledge their diagnostic errors because mistakes may be perceived as professional lapses or personal failings [46]. This practice may be one of the largest compromises of good health-care delivery.

## Study limitations

We could not conduct a more comprehensive retrospective review study because most hospital records of patients treated at the facility before July 2019 and beyond October 2019 could not be traced due to poor record-keeping, which requires improvement. Our results could be skewed by the large variations in the number of patients referred from the different provinces. 'Queried' diagnoses assumed 'definite' may have skewed the results of this study as in some cases, the final diagnosis may be different from the queried one. Also, since we only considered the primary diagnoses for our analysis, there might have been secondary referral diagnoses or complications that might have been correct but not considered. In addition, pre-referral medical treatment and possible delays in patients reaching the specialist ENT facility might have altered the disease process and resulted in perceived misdiagnoses. Further, the proportions of some of the diseases described in this study may vary with changing seasons. For instance, in Zambia, viral rhinopharyngitis and otitis media peak in the cold season. However, in our study, the prevalence of viral rhinopharyngitis and otitis media likely represented the annual incidence because the study period included both cold and hot months.

We acknowledge that our interpretation of 'inappropriate referral', even though according to Blundel et al. [17], was subjective and prone to bias as we relied on patient clinical files and referral documents for categorisation as appropriate or inappropriate. While it is a requirement in Zambia for clinicians to record investigations done and treatment instituted on the referral letter, some referring clinicians may not have inputted this information for the ENT specialist to note. Possibly, some of the reasons for not performing tests (i.e., non-functioning equipment, lack of reagents, unavailable expertise) at the referring facilities may not have been recorded on the referral letters, which may have increased the proportion of inappropriate referrals observed in our study. Another major limitation was that our utilisation of one ENT specialist to classify ENT diagnoses as matching or different was prone to bias.

Additional limitation to this study was the uncontrolled increase in familywise error rate across our statistical analyses. Overall, we consider this investigation relatively preliminary and encourage replication on an improved hospital record-keeping system that will achieve a longer study.

## Conclusions

In low-resource settings, more effective, accelerated ENT training of clinicians may improve the diagnostic accuracy and reduce the high non-specialist clinician diagnostic error and inappropriate patient referrals. Findings of this study may be used to guide policy aimed to improve ENT health care in Zambia.

## Supporting information

**S1 Appendix. Data collecting tool.**
(DOCX)

**S2 Appendix. The STROBE checklist for cross sectional studies.**
(DOCX)

**S1 Table. Proportional representation of ENT diseases treated at UTH ENT clinic.**
(DOCX)

**S2 Table. Misdiagnosis across age groups, ENT subspecialties, referral facilities level of care and provinces.**
(DOCX)

**S3 Table. Appropriateness of referral across age groups, ENT subspecialties, referral facilities level of care and provinces.**
(DOCX)

## Acknowledgments

We would like to thank the management of the University Teaching Hospitals (UTH) for their permission to conduct the research. Our gratitude also goes to Kasoka Lukama and Chabota Mungambata for transferring the patient information hard copies to the excel spreadsheet. Further, we would like to thank all members of staff at the UTH ENT clinic for helping to retrieve the patient records for the study. In addition, we would like to thank the EDCTP-TESA II/UTH Project in Zambia for sponsoring Lufunda Lukama to their 2019 Data Analysis and Manuscript Writing Bootcamp in Livingstone, Zambia.

## Author Contributions

**Conceptualization:** Lufunda Lukama, Colleen Aldous.

**Formal analysis:** Lufunda Lukama, Chester Kalinda.

**Investigation:** Lufunda Lukama.

**Methodology:** Lufunda Lukama.

**Resources:** Lufunda Lukama.

**Supervision:** Colleen Aldous, Charles Michelo, Chester Kalinda.

**Writing – original draft:** Lufunda Lukama.

**Writing – review & editing:** Colleen Aldous, Charles Michelo, Chester Kalinda.

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
