## [Decision Letter · Decision Letter 0]

6 Nov 2022

PONE-D-22-27161Ear, Nose and Throat (ENT) disease diagnostic error in low-resource health care: observations from a hospital-based cross-sectional studyPLOS ONE

Dear Dr. Lukama,

Thank you for submitting your manuscript to PLOS ONE. After careful consideration, we feel that it has merit but does not fully meet PLOS ONE’s publication criteria as it currently stands. Therefore, we invite you to submit a revised version of the manuscript that addresses the points raised during the review process.

We look forward to receiving your revised manuscript.

Kind regards,

Jorge Spratley, MD, PhD

Academic Editor

PLOS ONE

Journal Requirements:

"The authors received no specific funding for this work. However, Lufunda Lukama received financial support from the University of KwaZulu-Natal through his university tuition fee waiver and from the Canon Collins Educational and Legal Assistance Trust through the scholarship offered to him to study his Doctor of Philosophy in Medicine."

**Additional Editor Comments:**

Dear Authors,

The topic of your research is interesting. Please address carefully all the constructive questions/comments posed by the referees.

Reviewers' comments:

Reviewer's Responses to Questions

**Comments to the Author**

1. Is the manuscript technically sound, and do the data support the conclusions?

Reviewer #1: Partly

Reviewer #2: Yes

Reviewer #3: Yes

2. Has the statistical analysis been performed appropriately and rigorously? 

Reviewer #1: Yes

Reviewer #2: Yes

Reviewer #3: Yes

3. Have the authors made all data underlying the findings in their manuscript fully available?

Reviewer #1: Yes

Reviewer #2: Yes

Reviewer #3: Yes

4. Is the manuscript presented in an intelligible fashion and written in standard English?

Reviewer #1: Yes

Reviewer #2: Yes

Reviewer #3: Yes

5. Review Comments to the Author

Reviewer #1: It’s a well written manuscript, on a fluent English. It addresses a verry acute problem in Africa – the lack of specialists. Therefore, it merits our full attention.

The author pretends to demonstrate a high rate of diagnosis mismatch between the referring practitioner and the otolaryngologist, as well as high rate of inappropriate referral. However, we must point some critical biases that are not discussed and that might lower these rates.

L 74 The authors claim that “a more comprehensive retrospective review was not practical as most of the patients’ records from July 2019 and beyond October 2019 could not be traced due to poor record-keeping” and says it’s common in Zambia, citing another paper. This other paper’s authors managed however to collect data from earlier years, and so we don’t find it credible that only 4 months could be used. And a short period of 4 winter months is surely introducing biases. (And please review the confusing phrase – “prior to” and “more recent than”.

L84 “in the absence of…” this is a limitation in the author’s Hospital, or in the referring Hospitals? (If it’s in the referring Hospital, we would consider it acceptable, for as pointed before, only the UTH has ENT specialists. Also, it is a very specific issue: why mention this one limitation, and in relation to the pharynx alone? Why not mention laryngeal endoscopy? Or list all other limitations… I suggest you clarify why this issue is specifically relevant, or else remove it.

L99 Was the considered ENT diagnosis the first impression diagnosis or the diagnosis after tests? Would you consider a misdiagnose a diagnose that was not possible because of lack of access to tests? (For instance, a diagnosis of tracheal tumor instead of laryngeal papilloma in a Hospital with no endoscopy).

L104 would mean “all available primary care options had not been exhausted”?

112 “referral reason” is “referral diagnosis”?

L132 The author state that “Informed consent from the patients was not necessary and, therefore, was not obtained”. Consent is always needed, but under certain circumstances may be dismissed. So please address this issue here, admit it was not obtained because of some relevant reason (suggest e.g the research is of minimal risk, methods for protecting confidentiality are well outlined, and identifiers are destroyed as soon as possible…) and this was known to the Ethics Committee when presented for approval.

L 143 If 11% were self-referrals, who made the referral diagnosis? If the self-referred were self-diagnosed too, they shouldn’t be included. It seems you considered this latter in L172 but not here.

L153 “most misdiagnosis were from level 3 hospitals” may lead to incorrect interpretation, for these hospitals were the ones referring the most in total. In fact, they were in proportion more accurate than level 1, and we can assume that they are even leveled with the other two levels. So, the authors construction is misleading.

L171 the total number (591) does not match the number of included participants

L186 1374 referrals does not agree to L138 1543 referrals. Please correct it.

L330 Limitations’ discussion: you might discuss that some referral diagnosis might be a misdiagnosis because of time delay after referral? (e.g., tonsillar hypertrophy that got better) or just upgraded or downgraded (sinusitis to allergic rhinitis, or vice versa – allergic rhinitis often has sinusitis). We are never told how long it took from referral to ENT consulting, and neither if treatment was done in the meantime while waiting. The patient might have been referred for allergic rhinitis in the spring and was observed by the ENT in August presenting with a cold with sinusitis, for instance.

Also, the patient might have more than one diagnosis and at the time of referral - the most relevant one might not be that same when seen by the ENT surgeon (we must consider this when chronic sinusitis is mislabeled as recurrent tonsillitis, would you agree it’s hard to confuse on with the other?).

Lack of endoscopy or other tests might also be responsible (tracheal tumor in a case of laryngeal papilloma is a fair mistake, one might say, and in any case deserving referral). And finally, treatment (as you state in L188 for 25%) may have changed the initial diagnosis.

Also, referral appropriateness limitations are not discussed. The tools used to classify appropriateness pointed by the authors (as much as by Blundel) are highly subjective, and prone to misjudgment. For instance, were the authors able to retrospectively verify that all the needed tests were indeed available (and wrongly not used) at the time of referral? Was this done for every hospital? Can we confirm that all the primary care options were indeed available at that time, on a retrospective study? I know (having myself worked in Africa) that such availability varies a lot, and machines that are supposed to be working, sometimes have malfunctions that take a very long time to fix.

And finally, how do referral appropriateness and misdiagnosis relate to each other? Inappropriate referrals are often misdiagnosis too?

The authors write a long discussion text that leans on political and economy options and suggestions. I find that this part of the discussion is too long, and in part grows far from the studied subject, drawing some assumptions that are not supported or not related to the presented results. However, they leave out one suggestion that has been working very well in solving the referral issues in some other African countries, that is telemedicine.

In conclusion, I consider this manuscript a relevant one, but I urge the authors to reconsider some of the labeled misdiagnosis, and to address and solve the pointed biases, namely extend the studied records for the rest of the year.

Reviewer #2: Manuscript Number PONE-D-22-27161

This study is of interest for publication in this journal. The study presents an analysis of the diagnostic accuracy in patients with ENT pathology and referred to a tertiary hospital.

Major revisions

1. Authors performed an evaluation of the diagnostic accuracy of the patients refereed to another hospital and identified the existence of misdiagnosis cases. However, the authors should not indicate that the objective is the evaluation of misdiagnosis because the real number of misdiagnosis will be much higher. This analysis does not allow the identification of patients who initially had a wrong diagnosis and a wrong treatment but improved without the need to be referred to another hospital. Patients who received a curative treatment, despite the wrong diagnosis, are also not included (example: patient with bacterial tonsillitis with otalgia and medicated with antibiotic therapy for the otitis (wrong diagnosis) but improves with treatment because antibiotics cured the tonsillitis. Thus, this work does not assess misdiagnosis but assesses diagnostic accuracy in patients referred to another hospital. Thus, the objective of the study should be changed.

Minor revisions

1. ENT is a medical and surgical specialty. Thus, "ENT surgeon" should be replaced by "ENT doctor", "ENT specialist" or “ENT clinician”.

2. Authors performed an evaluation between 1 July 2019 and 30 October 2019 because poor record-keeping. However the authors do not explain why there were good clinical records in this period. If these records were more detailed due to the existence of the study, the choice of this time period must be explained, and related to the seasonality of some ENT pathologies;

3. Line 153 - “Despite non-ENT clinicians correctly diagnosing 72.0% (n=18)….” Authors don’t know the percentage of correct diagnosis, the authors only know the number of patients referred with a correct diagnosis and don’t know the correct diagnosis non-refereed.

4. Table 1 – I don’t understand the reason to refer an acute tonsillitis to a tertiary hospital (recurrent tonsillitis to surgery or tonsillitis resistant to treatment?). The same to allergic rhinitis and acute otitis media. This should be explained in the methods and could be important in understanding the diagnostic accuracy in the discussion.

5. The author should reinforce the comparison with developed countries (add comparison with a European country) in percentage of diagnostic accuracy.

Reviewer #3: After reviewing manuscript number PONE-D-22-27161 Research Article, with Title Ear, Nose and Throat (ENT) disease diagnostic error in low-resource health care: observations from a hospital-based cross-sectional study, we found it to have a technically acceptable scientific basis. The presentation of the document is intelligible, clear and unequivocal scientific language was used, and the conclusions presented are based on the data collected. It is also understood that you have made the data underlying the manuscript, which is important not only for the readers but also for its scientific character, available without restrictions. For ethical reasons, it is important to avoid duplication of this publication

6. PLOS authors have the option to publish the peer review history of their article (what does this mean?). If published, this will include your full peer review and any attached files.

Reviewer #1: **Yes: **Joao Subtil

Reviewer #2: **Yes: **Jorge Rodrigues

Reviewer #3: **Yes: **Palmira Essenje Pintar Kuatoko, my name should appear as Kuatoko, Palmira, MD, MsC; Faculty of Medicin Agostinho Neto University Angola; Josina Machel Hospital Angola

---

## [Author Response · Author response to Decision Letter 0]

21 Dec 2022

Editor Comments

Editor comment:

Authors' response/action:

We thank the editors for reviewing the manuscript. We have reviewed PLOS ONE’s style requirements and submitted our manuscript accordingly.

Editor comment:

2. Please provide additional details regarding participant consent. In the ethics statement in the Methods and online submission information, please ensure that you have specified (1) whether consent was informed and (2) what type you obtained (for instance, written or verbal, and if verbal, how it was documented and witnessed). 

If your study included minors, state whether you obtained consent from parents or guardians. If the need for consent was waived by the ethics committee, please include this information

Authors' response/action:

Thank you for this important comment. The study was a retrospective cross-sectional review of clinical case records of patients. To include all the requested detail, we amended our Ethics statement to read: 

‘The protocol and procedures of this study were approved by the University of KwaZulu-Natal Biomedical Research Ethics Committee (Ref: BREC/00001984/2020) following the South African national guidelines on Biomedical Research. In addition, ethics approval was obtained from the University of Zambia Biomedical Research Ethics Committee (Ref: REF. NO. 1380-2020) and the Zambia National Health Research Authority (Ref: NHRA00007/15/03/2021). Informed consent was waived by the Ethics Boards as the research was of minimal risk to the patient. Gatekeeper permission to conduct the study was obtained from UTH [no Reference number]. All data was fully anonymised following completion of the study. The study was cleared for publication by the Zambia National Health Research Authority (NHRA).’

Edited sections [Page/Line]

Methods section, Ethics subsection, Page 6, Lines 130-139

Editor comment:

Authors' response/action:

We are grateful for this comment and acknowledge the importance of providing grant numbers In the manuscript. Unfortunately, we are unable to provide the grant numbers as both the University of KwaZulu-Natal (UKZN) and Canon Collins Educational and Legal Assistance Trust do not assign grant numbers to their tuition waivers and scholarships respectively. We have, however, provided websites for both institutions. For any clarifications, please contact the University of KwaZulu-Natal and Canon Collins on the following emails:

Lushenthree Konar, UKZN, Konar@ukzn.ac.za Eva Lenica, Canon Collins Trust, Eva@canoncollins.org

We have also amended our funding information to read: 

‘The authors received no specific funding for this work. However, during the course of this study, Lufunda Lukama received a tuition fee waiver from the University of KwaZulu-Natal (https://ukzn.ac.za) and a scholarship (tuition fees, living cost stipend) from the Canon Collins Educational and Legal Assistance Trust (https://canoncollins.org/) for his Doctor of Philosophy studies at the University of KwaZulu-Natal College of Health Sciences in South Africa [no grant numbers available]. There was no additional external funding received for this study. The funders had no role in study design, data collection and analysis, decision to publish, or preparation of the manuscript.’

Edited sections [Page/Line]

Funding Statement included in the cover as instructed by the editor

Editor comment:

"The authors received no specific funding for this work. However, Lufunda Lukama received financial support from the University of KwaZulu-Natal through his university tuition fee waiver and from the Canon Collins Educational and Legal Assistance Trust through the scholarship offered to him to study his Doctor of Philosophy in Medicine."

Authors' response/action:

We are grateful for this guidance. We have amended our financial statement to read:

‘The authors received no specific funding for this work. However, during the course of this study, Lufunda Lukama received a tuition fee waiver from the University of KwaZulu-Natal (https://ukzn.ac.za) and a scholarship (tuition fees, living cost stipend) from the Canon Collins Educational and Legal Assistance Trust (https://canoncollins.org/) for his Doctor of Philosophy studies at the University of KwaZulu-Natal College of Health Sciences in South Africa [no grant numbers available]. There was no additional external funding received for this study. The funders had no role in study design, data collection and analysis, decision to publish, or preparation of the manuscript.’

Unfortunately, we are unable to provide the grant numbers as both the University of KwaZulu-Natal (UKZN) and Canon Collins Educational and Legal Assistance Trust do not assign grant numbers to their tuition waivers and scholarships respectively. We have, however, provided websites for both institutions. For any clarifications, please contact the University and Canon Collins on the following emails:

Lushenthree Konar, UKZN, Konar@ukzn.ac.za Eva Lenica, Canon Collins Trust, Eva@canoncollins.org

Editor comment:

Authors' response/action:

Thank you for the guidance. We have included the amended financial statement in our cover letter.

Editor comment:

Authors' response/action:

We are grateful for this guidance and have read through http://journals.plos.org/plosone/s/data-availability#loc-unacceptable-data-access-restrictions for more information on unacceptable data access restrictions

a) There are no ethical or legal restrictions on sharing our de-identified data set

b) The data used in this study has been uploaded to a public site and can be accessed at Lukama, Lufunda; Aldous, Colleen; Michelo, Charles; Kalinda, Chester (2022), “Dataset on diagnostic errors of Ear, Nose and Throat (ENT) diseases ”, Mendeley Data, v1 http://dx.doi.org/10.17632/6tgz2db7yn.1

Additional Editor Comments:

Dear Authors,

The topic of your research is interesting. Please address carefully all the constructive questions/comments posed by the referees.

Authors' response/action:

We are grateful to the Editor and all the reviewers for their valuable constructive review of the manuscript and positive commentary. We have carefully addressed all their comments as reflecting below:

Editor comments:

Reviewer's comments:

Reviewer's Responses to Questions

Comments to the Author

Comment to author:

1. Is the manuscript technically sound, and do the data support the conclusions?

Reviewer #1: Partly 

Reviewer #2: Yes 

Reviewer #3: Yes

Authors' response/action:

We are grateful for this assessment of our manuscript

Comment to author:

2. Has the statistical analysis been performed appropriately and rigorously?

Reviewer #1: Yes 

Reviewer #2: Yes 

Reviewer #3: Yes

Authors' response/action:

We are grateful for this assessment of our manuscript

Comment to author:

3. Have the authors made all data underlying the findings in their manuscript fully available?

Reviewer #1: Yes 

Reviewer #2: Yes 

Reviewer #3: Yes

Authors' response/action:

We are grateful for this assessment of our manuscript

Comment to author:

4. Is the manuscript presented in an intelligible fashion and written in standard English?

Reviewer #1: Yes 

Reviewer #2: Yes 

Reviewer #3: Yes

Authors' response/action:

We are grateful for this assessment of our manuscript

5. Review Comments to the Author

Reviewer #1

Reviewer comment:

It’s a well written manuscript, on a fluent English. It addresses a verry acute problem in Africa – the lack of specialists. Therefore, it merits our full attention.

Authors' response/action:

We are glad the reviewer found it useful

Reviewer comment:

The author pretends to demonstrate a high rate of diagnosis mismatch between the referring practitioner and the otolaryngologist, as well as high rate of inappropriate referral. However, we must point some critical biases that are not discussed and that might lower these rates.

Authors' response/action:

We thank the reviewer for his observations

Reviewer comment:

L 74 The authors claim that “a more comprehensive retrospective review was not practical as most of the patients’ records from July 2019 and beyond October 2019 could not be traced due to poor record-keeping” and says it’s common in Zambia, citing another paper. This other paper’s authors managed however to collect data from earlier years, and so we don’t find it credible that only 4 months could be used. 

Authors' response/action:

We are glad that the reviewer observed that in the cited paper, the authors managed to trace a larger proportion of records from the Cancer Diseases Hospital (CDH), Zambia’s only Oncology Centre. The fundamental difference between CDH and the University Teaching Hospitals (UTH), where our study was done is that CDH has a patient electronic data-base which makes retrieval of patient records easier, as opposed to the manual record keeping at UTH. Due to the lack of electronic record keeping at UTH, some patients are not entered into the manual registers; others even carry their files home. This is a general problem in the country which many hospitals are seeking solutions to.

The ENT Specialist who replaced the one that left UTH in July 2019 introduced an efficient patient record filing system that was abandoned when he was transferred to another tertiary facility in October 2019. After he left, the ENT clinic reverted to its default filing system, with some patients taking their medical files home and some not entered into the book. Even when these patients were manually registered into a book, there was frequently critical identifying data that was missing. This is fundamentally the reason why it was difficult to trace files of patients that were treated before July 2019 and after October 2019. Unfortunately, this was something we could not do anything about, but we note it creates an opportunity to design better filing systems at the facility. In making this clearer, we added the following statement to the Methods section of the manuscript: 

‘The trial filing system used between July 2019 and October 2019 had improved patient record keeping in the absence of an electronic filing system’

Edited sections [Page/Line]

Methods section, Design and setting subsection, Page 4, Lines 75-77

In addition, under the Limitations section, we have added the words ‘which requires improvement’ to the sentence ‘We could not conduct a more comprehensive retrospective review study because most hospital records of patients treated at the facility before July 2019 and beyond October 2019 could not be traced due to poor record-keeping’. The sentence now reads ‘We could not conduct a more comprehensive retrospective review study because most hospital records of patients treated at the facility before July 2019 and beyond October 2019 could not be traced due to poor record-keeping, which requires improvement.’

Edited sections [Page/Line]

Discussion section, Study limitations subsection, Page 16, Line 341

Reviewer comment:

And a short period of 4 winter months is surely introducing biases. 

Authors' response/action:

Absolutely. We acknowledged this as a limitation in our originally submitted manuscript as ‘Further, the proportions of some of the diseases described in this study may vary with changing seasons. For instance, in Zambia, viral rhinopharyngitis and otitis media peaks in the cold season. However, in our study, the prevalence of viral rhinopharyngitis and otitis media likely represented the annual incidence because the study period included both cold and hot months’ [Discussion section, Study limitations subsection, Page 16, Lines 349-353]. We have gone further to state:

‘Overall, we consider this investigation relatively preliminary and encourage replication on an improved hospital record keeping system that will achieve a longer study.’

Edited sections [Page/Line]

Discussion section, Study limitations subsection, Page 17, Lines 366-368

Reviewer comment:

(And please review the confusing phrase – “prior to” and “more recent than”.

Authors response/action:

Thank you for this suggestion. We have amended the statement to read:

‘A more comprehensive retrospective review was not practical as most of the patients’ records that were supposed to be filed before July 2019 and later than October 2019 could not be traced due to poor record-keeping, a general problem noted in Zambia [14]. The trial filing system used between July 2019 and October 2019 improved patient record keeping in the absence of an electronic filing system.’

Edited section [Page/Line]

Methods section, Design and setting subsection, Pages 3-4, Lines 72-77

Reviewer comment:

 L84 “in the absence of…” this is a limitation in the author’s Hospital, or in the referring Hospitals? (If it’s in the referring Hospital, we would consider it acceptable, for as pointed before, only the UTH has ENT specialists. Also, it is a very specific issue: why mention this one limitation, and in relation to the pharynx alone? Why not mention laryngeal endoscopy? Or list all other limitations… I suggest you clarify why this issue is specifically relevant, or else remove it.

Authors' response/action:

We are grateful for this comment and thank the reviewer for their guidance. 

This limitation is in the referring hospitals. We do agree with the suggestion that we either list all the limitations or remove the one we have reported. As listing the limitations would significantly lengthen the manuscript, we have gone with the option of removing it. Accordingly, the sentence ‘In the absence of endoscopes, lateral soft tissue x-rays are utilised in the diagnosis of postnasal space masses (including enlarged adenoids)’ has been deleted from the Methods section

Edited section [Page/Line]

Sentence deleted from the Methods section 

Reviewer comment:

L99 Was the considered ENT diagnosis the first impression diagnosis or the diagnosis after tests? Would you consider a misdiagnose a diagnose that was not possible because of lack of access to tests? (For instance, a diagnosis of tracheal tumor instead of laryngeal papilloma in a Hospital with no endoscopy).

Author response/action:

We are grateful that the reviewer picked up this ambiguity.

The considered ENT diagnosis was the final diagnosis, that arrived at after investigations (tests, imaging, endoscopy, etc). The occasional definitive diagnoses that remained elusive after investigations (queried diagnoses) were assumed definite, as already stated in the Methods section as ‘Queried diagnoses made by the referring clinicians and the specialist ENT surgeon were assumed definite because they often fall within the broad category of the actual patient pathology, as has been the principal researcher’s experience.’ [Methods section, Measurements and outcomes subsection, Page 5, Lines 99-101]

For clarity, we have added the sentence ‘The considered specialist ENT surgeon diagnosis was that made after investigating the patient as applicable (definitive or final diagnosis)’ to the methods section of the manuscript.

Edited sections [Page/Line]

Methods section, Measurements and outcomes subsection, Pages 4-5, Lines 95-97

Reviewer comment:

L104 would mean “all available primary care options had not been exhausted”?

Authors' response/action:

Correct. Thank you for picking that up. We had omitted the word ‘not’, which has now been inserted.

Edited section [page/Line]

Methods section, Measurements and outcomes subsection, Page 5, Line 107

Reviewer comment:

112 “referral reason” is “referral diagnosis”?

Authors' response/action:

The two terms are different.

The referral diagnosis is the disease (e.g., ear wax impaction) and the reason for referral would be why the patient is being referred to the ENT surgeon and not managed at the local facility (e.g., for ear irrigation because of lack of ear syringe). Another example would be for the diagnosis of complicated sinusitis (referral diagnosis) sent to the specialist for cross-sectional imaging and sinus surgery (referral reason)

Reviewer comment:

L132 The author state that “Informed consent from the patients was not necessary and, therefore, was not obtained”. Consent is always needed, but under certain circumstances may be dismissed. So please address this issue here, admit it was not obtained because of some relevant reason (suggest e.g. the research is of minimal risk, methods for protecting confidentiality are well outlined, and identifiers are destroyed as soon as possible…) and this was known to the Ethics Committee when presented for approval.

Authors' response/action:

We thank the reviewer for this critical observation. We in fact did apply for an ethics waiver as the risk of the research to the patient was minimal, which was granted by the Ethics Boards. We have since amended our Ethics section of the manuscript to read:

‘The protocol and procedures of this study were approved by the University of KwaZulu-Natal Biomedical Research Ethics Committee (Ref: BREC/00001984/2020) following the South African national guidelines on Biomedical Research. In addition, ethics approval was obtained from the University of Zambia Biomedical Research Ethics Committee (Ref: REF. NO. 1380-2020) and the Zambia National Health Research Authority (Ref: NHRA00007/15/03/2021). Informed consent was waived by the Ethics Boards as the research was of minimal risk to the patient. Gatekeeper permission to conduct the study was obtained from UTH [no Reference number]. All data was fully anonymised following completion of the study. The study was cleared for publication by the Zambia National Health Research Authority (NHRA).’

Edited section [Page/Line]

Methods section, Ethics subsection, Page 6, Lines 130-139

Reviewer comment:

L 143 If 11% were self-referrals, who made the referral diagnosis? If the self-referred were self-diagnosed too, they shouldn’t be included. It seems you considered this latter in L172 but not here

Authors' response/action:

Thank you for this enquiry.

Self-referrals did not have a referral diagnosis and, as the reviewer has rightly pointed out, were therefore not included in analysis of diagnosis mismatch, appropriateness of referrals and diagnosis agreement (Results and analysis section, Pages 7-10, Lines 152-204). The ENT disease epidemiology was determined from the diagnoses made by the ENT specialist, because it reflected the true incidence treated at the specialist centre (Results and analysis section, Page 7, Lines 147-151), Self-referrals were also given a diagnosis by the ENT specialist and excluding them from the determination of disease epidemiology would have been erroneous.

Therefore, we did not act on this comment.

Reviewer comment:

L153 “most misdiagnosis were from level 3 hospitals” may lead to incorrect interpretation, for these hospitals were the ones referring the most in total. In fact, they were in proportion more accurate than level 1, and we can assume that they are even leveled with the other two levels. So, the authors construction is misleading

Authors' response/action:

We are grateful the reviewer noticed this error. We have corrected that statement to read ‘Most (74.1%, n=80) misdiagnoses were from level 1 hospitals (S2 Table).’ 

The discussion section relating to this correction has not been altered as the facts recorded in that section are accurate.

Edited section [Page/Line]

Results and analysis section, Diagnostic accuracy and mismatch subsection, Page 7, Lines 156-157

Reviewer comment:

L171 the total number (591) does not match the number of included participants

Authors' response/action:

The number in Table 1 does not match the number of included participants because, for the purpose of this manuscript, we only included the 10 most common diagnoses made at the specialist facility, and noted it in our statement ‘Table 1 outlines the common diagnoses made by the ENT specialist at UTH ENT clinic, and proportions of their corresponding referring (non-ENT) clinician diagnoses’ contained in the Results and analysis section, Diagnosis accuracy and mismatch subsection, Page 7, Lines 160-161. We felt including all the many diagnoses made would make the table unnecessarily long and add little value to the manuscript. 

We therefore did not act on this suggestion.

Reviewer comment:

L186 1374 referrals does not agree to L138 1543 referrals. Please correct it.

Authors' response/action:

We acknowledge this suggestion and apologise for not making this clear enough, and for the erroneous figure 1374 instead of 1372.

1543 were all patients included in the study, stated as ‘1543 patients of age 0-87 years (mean age 25 � 21 years [mean �SD]) were included in the study, comprising 47.3% (n = 729) children (�18 years of age) and 52.7% (n=814) adults’ in our submitted manuscript [Results and analysis section, Patient demographics subsection, Page 6, Lines 142-143]. The 1372 referrals were those from non-ENT specialist clinicians (1543 total patients less 171 self-referrals).

To remove the ambiguity, under the Results and analysis section, we have amended the sentence ‘Of the total 1374 referrals, 56.6% (n=777) were inappropriate’ to read ‘Of the total 1372 non-ENT specialist clinician referrals, 56.6% (n=777) were inappropriate’. 

Edited section [Page/Line]

Results and analysis section, Appropriateness of referrals subsection, Page 9, Line 185

Reviewer comment:

L330 Limitations’ discussion: you might discuss that some referral diagnosis might be a misdiagnosis because of time delay after referral? (e.g., tonsillar hypertrophy that got better) or just upgraded or downgraded (sinusitis to allergic rhinitis, or vice versa – allergic rhinitis often has sinusitis). We are never told how long it took from referral to ENT consulting, and neither if treatment was done in the meantime while waiting. The patient might have been referred for allergic rhinitis in the spring and was observed by the ENT in August presenting with a cold with sinusitis, for instance.

Authors response/action:

Thank you for this important suggestion. 

We do acknowledge that delays in patients reaching the specialist are not uncommon in Zambia. However, it is very rare to get a patient referral letter without the date of referral (the date the letter was written) and unusual to have the treatment instituted not recorded on the referral letter. Therefore, the reviewers’ view that ‘We are never told how long it took from referral to ENT consulting, and neither if treatment was done in the meantime while waiting’ may be inaccurate if applied to the Zambian setting. The date of referral and treatment started is nearly always reflected on the referral letter, so delays would be extrapolated from that information.

We have acted on this suggestion by adding the sentence ‘In addition, pre-referral medical treatment and possible delays in patients reaching the specialist ENT facility might have altered the disease process and resulted in perceived misdiagnoses’ to our Study limitations section of the discussion.

Edited section [Page/Line]

Discussions section, Study limitations subsection, Page 16, Lines 346-348

Reviewer comment:

Also, the patient might have more than one diagnosis and at the time of referral - the most relevant one might not be that same when seen by the ENT surgeon (we must consider this when chronic sinusitis is mislabeled as recurrent tonsillitis, would you agree it’s hard to confuse on with the other?).

Authors' response/action:

A valid observation, thank you. We already acknowledged this fact in our Study limitations subsection of the Discussion section as ‘Also, since we only considered the primary diagnoses for our analysis, there might have been secondary referral diagnoses or complications that might have been correct but not considered’ [Lines 344-346]. In dealing with this further, we have added the sentence ‘In addition, pre-referral medical treatment and possible delays in patients reaching the specialist ENT facility might have altered the disease process and resulted in perceived misdiagnoses’ to our limitations section of the discussion’.

It is indeed hard to confuse chronic sinusitis with recurrent tonsillitis, and part of the explanation could be what the reviewer has suggested (addressed as above). However, this explanation may not apply to all cases, and we do see some surprising misdiagnoses, as in this case!

Edited section [Page/Line]

Discussions section, Study limitations subsection, Page 16, Lines 346-348

Reviewer comment:

Lack of endoscopy or other tests might also be responsible (tracheal tumor in a case of laryngeal papilloma is a fair mistake, one might say, and in any case deserving referral). 

Authors' response/action:

True. The lack of endoscopy or other tests as a cause of misdiagnosis was acknowledged in our submitted manuscript, stated as:

‘However, the widespread shortage of ENT diagnostic instruments across Zambia may explain the higher likelihood of misdiagnosis of patients with ear and those with sinonasal diseases. As nasal specula, otoscopes and endoscopes are seldom available in health facilities [5], clinicians seldomly perform complete ear and nasal examinations for diagnostic purposes. Except that of the larynx, much of the head and neck pathology requires no instrumentation to complete. In addition to the lack of sufficient skills and equipment at lower-level hospitals, patient referral networks in Zambia and many other low-income nations are undermined by poor coordination between lower and upper-level hospitals [36]. This may, in part, explain the high rate of inappropriate referrals noted in this study’ (Discussion section, Page 13, Lines 273-281).

We also note that, while a referral for a tracheal tumour confused for laryngeal papilloma is a misdiagnosis, it is certainly an appropriate and deserving referral and was treated as such in our analysis. We are also of the view that, in the absence of endoscopic equipment, simple imaging e.g., x-ray may be successfully employed to make the diagnosis of tracheal or laryngeal tumours, coupled with good history taking and clinical examination.

Reviewer comment:

And finally, treatment (as you state in L188 for 25%) may have changed the initial diagnosis.

Authors 'response/action:

The 25% of inappropriate referrals referred to here are those that required trial of treatment before referral to ENT but did not get it. The patients sent for specialist care after treatment were those who did not respond to treatment, or partially responded to it, and it would be unlikely for the patient to present to the ENT with a different diagnosis.

However, we acknowledge this possibility and added ‘In addition, pre-referral medical treatment and possible delays in patients reaching the specialist ENT facility might have altered the disease process and resulted in perceived misdiagnoses’ to our limitations section of the discussion.

Edited section [[Page/Line]

Discussion section, Study limitations subsection, Page 16, Lines 346-348

Reviewer comment:

Also, referral appropriateness limitations are not discussed. The tools used to classify appropriateness pointed by the authors (as much as by Blundel) are highly subjective, and prone to misjudgment. For instance, were the authors able to retrospectively verify that all the needed tests were indeed available (and wrongly not used) at the time of referral? Was this done for every hospital? Can we confirm that all the primary care options were indeed available at that time, on a retrospective study? I know (having myself worked in Africa) that such availability varies a lot, and machines that are supposed to be working, sometimes have malfunctions that take a very long time to fix.

Authors' response/action

We thank the author for this critical observation. We have addressed this concern by adding the following section to the limitations section of the manuscript:

‘We acknowledge that our interpretation of ‘inappropriate referral’, even though according to Blundel et al. [17], was subjective and prone to bias as we relied on patient clinical files and referral documents for categorisation as appropriate or inappropriate. While it is a requirement in Zambia for clinicians to record investigations done and treatment instituted on the referral letter, some referring clinicians may not have inputted this information for the ENT specialist to note. Possibly, some of the reasons for not performing tests (i.e., non-functioning equipment, lack of reagents, unavailable expertise) at the referring facilities may not have been recorded on the referral letters, which may have increased the proportion of inappropriate referrals observed in our study. Another major limitation was that our utilisation of one ENT specialist to classify ENT diagnoses as matching or different was prone to bias.’ 

Edited section [Page/Line]

Discussion section, Limitations subsection, Pages 16-17, Lines 354-364

Reviewer comment:

And finally, how do referral appropriateness and misdiagnosis relate to each other? Inappropriate referrals are often misdiagnosis too?

Authors' response/action:

For this study, we defined ‘inappropriate referral’ according to Blundel et al. [17]. A referral was inappropriate if it was either unnecessary (all available primary care options had not been exhausted), and/or had a wrong referral destination (patient referred to ENT instead of another medical speciality), and/or was of poor quality (necessary tests and investigations were not performed considering the available resources of the referring facility). Misdiagnosis is an ‘erroneous diagnosis’ [1]; the patient receives an incorrect diagnosis. While misdiagnoses may lead to inappropriate referrals, inappropriate referrals may not result in misdiagnosis.

1. Dong, D., Chung, R.YN., Chan, R.H.W. et al. Why is misdiagnosis more likely among some people with rare diseases than others? Insights from a population-based cross-sectional study in China. Orphanet J Rare Dis 15, 307 (2020). https://doi.org/10.1186/s13023-020-01587-

Reviewer comment:

The authors write a long discussion text that leans on political and economy options and suggestions. I find that this part of the discussion is too long, and in part grows far from the studied subject, drawing some assumptions that are not supported or not related to the presented results. However, they leave out one suggestion that has been working very well in solving the referral issues in some other African countries, that is telemedicine

Authors' response/action:

We thank the reviewer for this excellent suggestion and note their concern. We recognise, too, that while telemedicine offers a great practical solution in the improvement of health services, it requires political will and funding for its establishment and sustainability, especially in developing countries like Zambia. In addressing the concern raised , we have added the following section to the discussion of our manuscript:

‘Further, the introduction of ENT telemedicine to Zambia may improve patients’ access to quality health care. While telemedicine has been successfully implemented in other medical subspecialties in Zambia [41], it is yet to be used in ENT health care. However, to ensure success of telemedicine, the government must develop its own implementation strategies that are tandem with Zambia’s needs [42].

Edited section [Page/Line]

Discussion section, Page 15, Lines 318-322

Reviewer comment:

In conclusion, I consider this manuscript a relevant one, but I urge the authors to reconsider some of the labeled misdiagnosis, and to address and solve the pointed biases, namely extend the studied records for the rest of the year. 

Authors' response/action

We are glad the reviewer considered our manuscript relevant. We have addressed the issue of some of the labelled misdiagnoses and biases in our responses above. Unfortunately, we are unable to extend our retrospective study due the poor record keeping at the facility. The explanation is as below:

We are also appreciative of the fact that the reviewer observed that in the cited paper, the authors managed to trace more records from the Cancer Diseases Hospital (CDH), Zambia’s only Oncology Centre. The fundamental difference between CDH and the University Teaching Hospitals (UTH), where our study was done is that CDH has a patient electronic data base which makes retrieval of patient records easier, as opposed to the manual record keeping at UTH. Due to the lack of electronic record keeping at UTH, some patients are not entered into the manual registers, others carry their files home. This is a general problem in Zambia and other resource-limited nations. The lack of electronic record keeping systems as a reason for the sparsity of data relating to diagnostic error in poor countries was referenced in our manuscript’s introduction as ’However, in low-resource settings, data relating to harmful diagnostic error remains sparse due to limited access to diagnostic resources, shortage of qualified medical professionals and poverty of electronic record-keeping systems [11]’ [Introduction section, page 3, Lines 56-59]

(Reference 11: Singh H, Schiff GD, Graber ML, Onakpoya I, Thompson MJ. The global burden of diagnostic errors in primary care. BMJ Qual Saf. 2017 Jun 1;26(6):484–94. )

The ENT Surgeon who replaced the one that left UTH in July 2019 introduced an efficient patient record filing system that was abandoned when he was transferred to another tertiary facility in October 2019. After he left, the ENT clinic reverted to its default filing system, with some patients taking their medical files home and some not entered into the book. Even when these patients were manually registered into a book, there was frequently critical identifying data that was missing. This is fundamentally the reason why it was difficult to trace files of patients that were treated before July 2019 and after October 2019. Unfortunately, this was something we could not do anything about, but creates an opportunity to design better filing systems at the facility. In making this clearer, we added the statement ‘The trial filing system used between July 2019 and October 2019 had improved patient record keeping in the absence of an electronic filing system’ to the Methods section of the manuscript [Methods section, Design and setting subsection, Pages 4, Lines 75-77]. In addition, under the Limitations section, we have added the words ‘, which requires improvement’ to the sentence ‘We could not conduct a more comprehensive retrospective review study because most hospital records of patients treated at the facility before July 2019 and beyond October 2019 could not be traced due to poor record-keeping.’ [Discussion section, Study Limitations subsection, Page 16, Line 341]

In acknowledging that a longer, more robust study is required, we included the section ‘Overall, we consider this investigation relatively preliminary and encourage replication on an improved hospital record keeping system that will achieve a longer study’ to our limitations section of the manuscript. [Discussion section, Study Limitations subsection, Page 17, Lines 366-368]

Edited section [Page/Line]

Methods section, Design and setting subsection, Page 4, Lines 75-77

Discussion section, Study Limitations subsection, Page 16, Line 341

Discussion section, Study Limitations subsection, Page 17, Lines 366-368

Reviewer #2: Manuscript Number PONE-D-22-27161

Reviewer comment:

This study is of interest for publication in this journal. The study presents an analysis of the diagnostic accuracy in patients with ENT pathology and referred to a tertiary hospital.

Authors' response/action:

We thank the reviewer for their accurate assessment of our manuscript.

Major revisions

Reviewer comment:

1. Authors performed an evaluation of the diagnostic accuracy of the patients refereed to another hospital and identified the existence of misdiagnosis cases. However, the authors should not indicate that the objective is the evaluation of misdiagnosis because the real number of misdiagnosis will be much higher. This analysis does not allow the identification of patients who initially had a wrong diagnosis and a wrong treatment but improved without the need to be referred to another hospital. Patients who received a curative treatment, despite the wrong diagnosis, are also not included (example: patient with bacterial tonsillitis with otalgia and medicated with antibiotic therapy for the otitis (wrong diagnosis) but improves with treatment because antibiotics cured the tonsillitis. Thus, this work does not assess misdiagnosis but assesses diagnostic accuracy in patients referred to another hospital. Thus, the objective of the study should be changed.

Authors' response/action:

An excellent observation. In addressing this concern, our initial objective ‘In this study, we estimated the prevalence of misdiagnosis and inappropriate referrals among patients treated at Zambia’s highest ENT treatment facility. In addition, we determined the level of agreement of ENT diagnosis between the specialist ENT surgeon and referring clinicians’ was restated:

‘In this study, we determined the diagnostic accuracy and estimated the appropriateness of referrals among patients treated at Zambia’s highest ENT treatment facility. In addition, we determined the level of agreement of ENT diagnosis between the ENT specialist and referring clinicians.’

Edited section [Page/Line]

Introduction section, Page 3, Lines 64-67

Minor revisions

Reviewer comment:

1. ENT is a medical and surgical specialty. Thus, "ENT surgeon" should be replaced by "ENT doctor", "ENT specialist" or “ENT clinician”.

Authors' response/action:

We have replaced ‘ENT surgeon’ with ‘ENT specialist’ throughout the manuscript

Edited section [Page/Line]

Lines 32, 66, 80, 95, 100, 102, 103, 115, 119, 126, 153, 160, 170, 173, 201, 202, 203, 204, 251, 312

Reviewer comment:

2. Authors performed an evaluation between 1 July 2019 and 30 October 2019 because poor record-keeping. However the authors do not explain why there were good clinical records in this period. If these records were more detailed due to the existence of the study, the choice of this time period must be explained, and related to the seasonality of some ENT pathologies;

Authors' response/action:

We are grateful for this comment and acknowledge the need to elaborate this further.

In our manuscript we cited a paper by Mumba et al. [1] that had similar difficulties in retrieving patient records. The study was done at Cancer Diseases Hospital (CDH), Zambia’s only Oncology Centre. In that study, however, the authors managed to trace more records because the hospital uses an electronic record keeping system. The fundamental difference between CDH and the University Teaching Hospitals (UTH), where our study was done is that CDH has a patient electronic data base which makes retrieval of patient records easier, as opposed to the manual record keeping at UTH. Due to the lack of electronic record keeping at UTH, some patients are not entered into the manual registers, others carry their files home. This is a general problem in Zambia and other resource-limited nations. The lack of electronic record keeping systems as a reason for the sparsity of data relating to diagnostic error in poor countries was referenced in our manuscript’s introduction as ’However, in low-resource settings, data relating to harmful diagnostic error remains sparse due to limited access to diagnostic resources, shortage of qualified medical professionals and poverty of electronic record-keeping systems [11]’ [Introduction section, page 3, Lines 57-59]

(Reference 11: Singh H, Schiff GD, Graber ML, Onakpoya I, Thompson MJ. The global burden of diagnostic errors in primary care. BMJ Qual Saf. 2017 Jun 1;26(6):484–94.)

The ENT Surgeon who replaced the one that left UTH in July 2019 introduced an efficient patient record filing system that was abandoned when he was transferred to another tertiary facility in October 2019. After he left, the ENT clinic slid right back into its default filing system, with some patients taking their medical files home and some not entered into the book. Even when these patients were manually registered into a book, there was frequently critical identifying data that was missing. This is fundamentally the reason why it was difficult to trace files of patients that were treated before July 2019 and after October 2019. Unfortunately, this was something we could not do anything about, but creates an opportunity to design better filing systems at the facility. In making this clearer, we added the statement ‘The trial filing system used between July 2019 and October 2019 had improved patient record keeping in the absence of an electronic filing system’to the Methods section of the manuscript [Methods section, Design and setting subsection, Pages 4, Lines 75-77]. In addition, we have rephrased part of the limitations section to read ‘We could not conduct a more comprehensive retrospective review study because most hospital records of patients treated at the facility before July 2019 and beyond October 2019 could not be traced due to poor record-keeping, which requires improvement’ [Discussion section, Study Limitations subsection, Page 16, Line 339-341]

The limitation relating to the seasonality of some of the ENT diseases was acknowledged in our manuscript as ‘Further, the proportions of some of the diseases described in this study may vary with changing seasons. For instance, in Zambia, viral rhinopharyngitis and otitis media peaks in the cold season. However, in our study, the prevalence of viral rhinopharyngitis and otitis media likely represented the annual incidence because the study period included both cold and hot months.’ [Discussion section, Study limitations subsection, page 16, Lines 349-353].

In acknowledging that a longer, more robust study is required, we included the section ‘Overall, we consider this investigation relatively preliminary and encourage replication on an improved hospital record keeping system that will achieve a longer study’ to our limitations section of the manuscript. [Discussion section, Study limitations subsection, page 17, Lines 366-368]

1. Mumba JM, Kasonka L, Owiti OB, Andrew J, Lubeya MK, Lukama L, et al. Cervical cancer diagnosis and treatment delays in the developing world: Evidence from a hospital-based study in Zambia. Gynecologic Oncology Reports. 2021 Aug 1;37:100784.

Edited sections [Page/Line]

Methods section, Design and setting subsection, Page 4, Lines 75-77

Discussion section, Study Limitations subsection, Page 16, Line 339-341

Discussion section, Study limitations subsection, page 17, Lines 366-368

Reviewer comment:

 3. Line 153 - “Despite non-ENT clinicians correctly diagnosing 72.0% (n=18)….” Authors don’t know the percentage of correct diagnosis, the authors only know the number of patients referred with a correct diagnosis and don’t know the correct diagnosis non-refereed

Authors' response/action:

We are glad the author picked this up. To make certain that the readers know that these figures apply to patients referred for specialist care and not to all patients (referred and non-referred) seen at the referring facility, we amended the applicable statement to read:

 ‘Despite non-ENT clinicians correctly diagnosing 72.0% (n=18) of nasal polyps and 51.5% (n=35) of acute tonsillitis in patients referred for specialist care, only 3.3% (n=4) of them correctly diagnosed allergic rhinitis.’

Edited section [Page/Line]

Results and analysis section, diagnostic accuracy and mismatch subsection, Page 7, Lines 157-159

Reviewer comment:

4. Table 1 – I don’t understand the reason to refer an acute tonsillitis to a tertiary hospital (recurrent tonsillitis to surgery or tonsillitis resistant to treatment?). The same to allergic rhinitis and acute otitis media. This should be explained in the methods and could be important in understanding the diagnostic accuracy in the discussion.

Authors' response/action:

We do agree with the author that some of the ENT conditions referred to the tertiary hospital should have been handled by the referring facilities. The ‘Reason for referral’ was captured in our data collection (see Methods section, Participant selection and sampling subsection, Page 4, Lines 90-92) and considered it for classifying the referral as appropriate or inappropriate according to Blundel et al. The discussion section below partly explained the possible reasons why patients with allergic rhinitis or acute tonsillitis may be referred to a tertiary hospital:

‘The poor agreement between the referring clinicians and ENT specialist (k=0.0001) and the high rate (56.6%) of inappropriate referrals for ENT specialist care may indicate inadequate training and poor knowledge of basic ENT disease management among most clinicians. In Zambia and the rest of the world, ENT is given little weight in medical school curricula, with most ENT undergraduate curricula offering a 1-2 week predominantly observational rotation [30]. As such, most clinicians lack the necessary skills to confidently manage ENT diseases upon completion of their undergraduate medical training [31]. Even if some literature emphasizes the superiority of adequate undergraduate ENT medical training over the duration spent as a general practitioner in the acquisition of competence to treat ENT conditions [32], the majority of clinicians often do require additional training to improve their competence following their undergraduate medical education [31]. Therefore, improving ENT medical education and clinical exposure at undergraduate, graduate and in-service training in Zambia would give clinicians better knowledge and skill to handle ENT conditions [33,32]. [Discussion section, Pages 12-13, Lines 251-263]

We are glad to report that we are currently conducting a study to gain more insight into why such referrals are prevalent. To show this, we have added the sentences:

‘A study assessing the knowledge, attitudes and current practices of health workers with regards to the basic management of ENT diseases in Zambia is ongoing.' [Discussion section, Page13, Lines 263-267] We hope that the study will provide more insight into why some uncomplicated and correctly diagnosed conditions (i.e., acute tonsillitis, allergic rhinitis) are sent for ENT specialist treatment.’

Edited sections [Page/Line]

Discussion section, Page13, Lines 263-267

Reviewer comment:

5. The author should reinforce the comparison with developed countries (add comparison with a European country) in percentage of diagnostic accuracy.

Authors' response/action:

We acknowledge this important suggestion. In response, we have added the sentence ‘Further, a study in the Netherlands reported diagnostic accuracy of 0.42 among medical residents, scored as either 0 (incorrect), 0.5 (partially correct) or 1 (correct)’ to the discussion section of the manuscript. The referenced paper has been inserted [reference 29]

Edited section [Page/Line]

Discussion section, Page 10, Lines 240-242

Reviewer #3

Reviewer comment:

After reviewing manuscript number PONE-D-22-27161 Research Article, with Title Ear, Nose and Throat (ENT) disease diagnostic error in low-resource health care: observations from a hospital-based cross-sectional study, we found it to have a technically acceptable scientific basis. The presentation of the document is intelligible, clear and unequivocal scientific language was used, and the conclusions presented are based on the data collected. It is also understood that you have made the data underlying the manuscript, which is important not only for the readers but also for its scientific character, available without restrictions. For ethical reasons, it is important to avoid duplication of this publication

Authors' response/action:

We are glad that the reviewer found that our manuscript had a technically acceptable scientific basis. We are also encouraged by their complements. 

We have made our data available without restrictions. We will not duplicate the publication.

Additional Changes to the manuscript

The STROBE checklist for cross sectional studies (Supplementary file 2) has been adjusted accordingly, with all changed page numbers highlighted in red. Also, with the addition of new references, we have updated the order of our references.

---

## [Editor Report · Decision Letter 1]

30 Jan 2023

Ear, Nose and Throat (ENT) disease diagnostic error in low-resource health care: observations from a hospital-based cross-sectional study

PONE-D-22-27161R1

Dear Dr. Lukama,

We’re pleased to inform you that your manuscript has been judged scientifically suitable for publication and will be formally accepted for publication once it meets all outstanding technical requirements.

Kind regards,

Jorge Spratley, MD, PhD

Academic Editor

PLOS ONE
---

## [Editor Report · Acceptance letter]

1 Feb 2023

PONE-D-22-27161R1 

Ear, Nose and Throat (ENT) disease diagnostic error in low-resource health care: observations from a hospital-based cross-sectional study 

Dear Dr. Lukama:

I'm pleased to inform you that your manuscript has been deemed suitable for publication in PLOS ONE. Congratulations! Your manuscript is now with our production department. 

Kind regards, 

on behalf of

Professor Jorge Spratley 

Academic Editor

PLOS ONE